# Iatrogenic Dementia: Providing Insight into Transmissible Subtype of Alzheimer’s Disease, Creutzfeldt–Jakob Disease and Cerebral Amyloid Angiopathy

**DOI:** 10.3390/biom15040522

**Published:** 2025-04-03

**Authors:** Stella Karatzetzou, Serafeim Ioannidis, Eleni Konstantinopoulou, Dimitrios Parisis, Theodora Afrantou, Panagiotis Ioannidis

**Affiliations:** 12nd Department of Neurology, AHEPA University Hospital, Aristotle University of Thessaloniki, 54636 Thessaloniki, Greece; skaratzetzou@gmail.com (S.K.); elinappg@yahoo.com (E.K.); dparisis@auth.gr (D.P.); afrantou@gmail.com (T.A.); 2School of Medicine, Aristotle University of Thessaloniki, 54124 Thessaloniki, Greece; ioannidissera@gmail.com

**Keywords:** Alzheimer’s disease, Creutzfeldt–Jakob disease, cerebral amyloid angiopathy, iatrogenic dementia, acquired dementia

## Abstract

Within the phenotypic spectrum of Alzheimer’s disease (AD), Creutzfeldt–Jakob disease (CJD) and cerebral amyloid angiopathy (CAA), dementia that is attributed to iatrogenic transmission has increasingly gained scientific attention recently. Newly recognized, this treatment-induced form of dementia may result from exposure to certain medical or surgical procedures. The present review aims to explore the distinct features of acquired dementia encompassing a history of potential exposure and relatively early age of onset, highlighting transmission potential with a rather prion-like pattern. Having reviewed all available relevant literature, dementia of iatrogenic etiology represents a new disease entity that requires an individualized investigation process and poses a great clinical challenge as far as patients with AD, CJD and CAA are concerned. Understanding the underlying pathophysiology of these rare forms of dementia may significantly enhance awareness within clinical field of neurodegenerative diseases and facilitate their prompt management.

## 1. Introduction

Neurodegenerative diseases represent a major public health concern in the context of a constantly ageing society, reflecting a noteworthy rise in the prevalence of such age-related disorders [1]. Encompassing a great variety of disease entities, Alzheimer’s disease (AD), Creutzfeldt–Jakob disease (CJD) and cerebral amyloid angiopathy (CAA) being among them [2], neurodegenerative diseases may manifest with a wide spectrum of clinical phenotypes. Cognitive decline, presenting early on the disease course or during a later stage of it, stands for a clinical key feature among patients suffering from the aforementioned disorders [3]. In a clinical setting of neurodegeneration, subtle cognitive deficits that progressively deteriorate with advanced age result in the development of clinically evident mild cognitive impairment (MCI), ultimately leading over time to dementia. Taking into account that according to the World Health Organization (WHO) dementia will affect 75 and 132 million people worldwide by the years 2030 and 2050, respectively, the overall dementia-associated socio-economic burden is expected to further increase in the future decades [4].

Thus, an emerging need is being highlighted regarding in-depth understanding of the related pathophysiological pathways and subsequently both accurate and prompt diagnostic approach, aiming at facilitating overall management and decision making within this group of patients.

As far as the pathophysiological basis of neurodegenerative disorders is concerned, most of them are attributed to intracellular or extracellular misfolding and aggregation of disease-specific proteins in the brain that are subsequently accumulated in the form of abnormal protein aggregates, capable of causing dysfunction and loss of neuronal cell population, like CJD [5,6,7]. As a result, a wide range of clinical symptoms is progressively being reported among patients with neurodegenerative disorders depending on the type of abnormally accumulated protein. Recently, an increasing body of evidence is supporting the fact that similar underlying mechanisms may be responsible for the replication and spread of distinct misfolded proteins preferentially within the central nervous system (CNS) [8]. Different disease-associated proteins, including amyloid-beta (Aß) and tau in AD, as well as amyloid-beta (Aß) in CAA, have been documented to share “prion-like” properties, mimicking several major traits of prions [9]. More specifically, it has been demonstrated that the ability to serve as a template for the conversion of a normally present brain protein in an abnormal misfolded protein aggregate that then propagates between cells and spreads across different anatomical pathways may characterize disease-causing proteins other than prions as well [10].

It is well established that prion diseases, naturally occurring in humans as well several animal species, may be transmitted across species barriers to other individuals. Noteworthy, CJD, the most common human prion disease, is characterized by a human-to-human transmission potential with a prolonged incubation period between underlying exposure and the onset of neurological symptoms [6]. Environmentally acquired CJD, including iatrogenic form, unlike sporadic and inherited subtypes, accounts only for a small proportion of all CJD cases with an incidence rate of less than 1% [11]. Although rare, iatrogenic CJD (iCJD) that is transmitted among individuals through both surgical and medical procedures, is of paramount importance in terms of public health. Taking into consideration the commonalities documented between CJD and other neurodegenerative diseases as far as their transmissibility nature and underlying pathogenic mechanisms are concerned, a number of concerns have inevitably been raised regarding if and to what extent AD and CAA exhibit similar properties in a clinical setting.

In recent years, iatrogenic dementia originating from an acquired neurodegenerative disease independently of its exposure source has emerged as a rather new and intriguing concept, pointing out potential public health implications. Garnering scientific interest and research attention lately, dementia of iatrogenic etiology poses a great clinical challenge as far as patients with AD, CJD and CAA are concerned. In an attempt to provide further valuable insights into the increasingly recognized transmissible type of cognitive decline, the objective of our study was to review all available literature published within the last decade dealing with the pathomechanisms, the distinct clinical features and transmission potential of acquired dementia.

## 2. Materials and Methods

The Preferred Reporting Items for Systematic Reviews and Meta-analyses (PRISMA checklist) was used to guide this study.

### Search Strategy-Selection Criteria

Literature research of two databases (MEDLINE and Scopus) was conducted in order to trace all relevant studies using either “iatrogenic dementia” as a keyword or related terms “transmissible dementia” or “treatment-induced dementia” or “acquired dementia” as search criteria. Moreover, the terms [“Creutzfeldt–Jakob disease”] OR [“Alzheimer’s disease”] OR [“cerebral amyloid angiopathy”] were used as second search criteria. The retrieved articles were also hand searched for any further potential eligible articles. Only full-text original articles published in the English language were included. Overall, 749 records were retrieved from the database search. Duplicates and irrelevant studies were excluded; hence, a total of 124 articles were selected. After screening the full text of the articles, 71 studies were eligible for inclusion (Figure 1).

## 3. Results

### 3.1. Human Prion Diseases

Iatrogenic Creutzfeldt–Jakob Disease (iCJD)

CJD, the main representative of human prion diseases, stands for a progressive, fatal neurodegenerative disorder characterized by a long latency and a rapidly evolving disease course with a rather short post-diagnosis survival [12]. Based on the prion hypothesis, CJD is attributed to the self-propagation, accumulation and spread of a misfolded disease-causing protein through anatomical pathways within the brain, leading to neuronal cell death and subsequently neurological dysfunction [13]. The vast majority of CJD cases occur as a sporadic form, followed by the genetic form of the disease, while environmentally acquired CJD, such as kuru, iatrogenic CJD, and variant CJD (vCJD), is coupled with a low incidence rate among all CJD cases [3,6]. The clinical phenotype varies among different disease subtypes including rapidly progressive dementia, cerebellar and extrapyramidal signs, myoclonus and visual symptoms [12].

Although rare, iatrogenic CJD is of significant concern to the wider population. Depending on the source of exposure, iCJD may be either surgical- or medical-associated. Several infection sources have been recognized, with cadaveric dura mater grafts (hDM) and human growth hormone (hGH) being the most commonly identified among CJD patients. Other potential transmission routes of the disease involve contaminated neurosurgical instruments, stereotactic electroencephalogram (EEG) electrodes, corneal transplants as well as human gonadotrophic hormone. Furthermore, transfusion of blood products may serve as a CJD transmission source, resulting in secondary infection with vCJD [3,6,11].

Although neurosurgical transmission of CJD had already been suggested in 1960 [14], the first case report was documented in a recipient of a cadaveric corneal transplant from a donor with pathologically proven CJD [15]. In the following years, further CJD cases have been reported to be associated with administration of contaminated corneal grafts and EEG recordings with deep needle electrodes [16,17,18], while in the mid-1980s hGH and hDM grafts were identified as exposure sources responsible for iCJD occurrence [19,20,21,22]. The incidence curves of hGH- and hDM-associated CJD exhibiting a broad peak occurred in the mid-to-late 1990s [11], when the implementation of both hGH and hDM grafts was ceased and replaced with recombinant pituitary hormones and dura substitutes (synthetic or derived from non-human animals), respectively. As a result of the prohibition of their broad administration, the two most predominant forms of iCJD have dramatically decreased [23], even though occasional cases may still occur due to the significantly lengthy incubation periods of human prion infections [6]. More recently, three cases of blood transfusion-related vCJD have been reported in recipients of non-leukodepleted red cell concentrates from asymptomatic donors who subsequently developed vCJD, suggesting an additional transmission route of particular concern [24].

With the iCJD exposure routes being well documented, iCJD cases recorded worldwide mostly occurred in recipients of contaminated hGH and hDM grafts derived from human cadavers with undiagnosed CJD infections. Until 2012, more than 450 CJD cases were attributed to the implementation of contaminated cadaveric dura mater (228 cases) and pituitary glands (226 cases) for neurosurgical procedures and hormone replacement therapy, respectively. As a result of alternative transmission routes, a small proportion of total CJD cases was identified up to 2012, encompassing utilization of neurosurgical instruments (four cases), electroencephalography recording using invasive medical devices (two cases), corneal transplants (two cases), gonadotropin hormone (four cases) and packed red blood cells (three cases) [11]. Since 2013, there have been only 16 additional iCJD cases reported [23].

Up to today, at least 485 iCJD cases have been reported worldwide with most of them (>96%) being identified before 2012, highlighting the fact that the case numbers and annual incidence of iCJD declined remarkably across the globe from 1993 to 2020 [23].

With regards to the clinical phenotype of iCJD, cases resulting from peripheral exposure to prions are mainly manifested with early ataxia of cerebellar onset and a more prolonged clinical course compared to iCJD associated with direct central nervous system exposure to prions typically coupled with cognitive symptoms. As a result, distinct clinical features are commonly observed among CJD patients depending on the source of prion exposure. More specifically, hGH-induced iCJD is supported to be coupled with a progressive cerebellar syndrome including gait ataxia, cerebellar dysarthria and lower limb pain, while cognitive function remains much less affected with only delayed dementia, if any [11,25]. Occurring at an unusually young age at onset, CJD, attributed to inter-human hGH prion transmission, is characterized by a development of sleep disturbance, cognitive decline with prominent memory decline and pyramidal signs with weakness in the lower limbs as late components of the clinical course [26]. Rarely, cognitive impairment emerges at an early stage [27]. On the contrary, hDM graft-associated iCJD becomes clinically evident based on the reported neuropathological findings, either as a plaque or a non-plaque type. Progressive, mainly ataxic, gait disturbance with a later development of myoclonus and akinetic mutism and a relatively slow disease progression represent atypical features that are present in the case of plaque type hDM graft-sourced iCJD, while the clinical profile of non-plaque type resembles that of typical sporadic CJD with an old age at onset and rapidly progressive dementia [25,28,29]. Interestingly, mean latency between prion transmission and disease onset is estimated to be around 17 and 12 years as far as hGH- and hDM graft-related iCJD are concerned, respectively [11].

### 3.2. Beyond Human Prion Diseases, Toward Other Neurodegenerative Diseases

According to recently accumulated evidence, specific disease-causing proteins representing the hallmark of various neurodegenerative diseases may mimic several major prion characteristics, Aß and tau being among them. As a result, misfolded and aggregated abnormal forms of the aforementioned proteins are capable of inducing neurodegeneration by a prion-like mechanism leading to common neurodegenerative diseases, including CAA and AD. Newly emerging data raise the question of whether human prion diseases serve as a model for a wide range of neurodegenerative disorders and if such disorders might exhibit a similar human-to-human transmission potential under certain circumstances.

#### 3.2.1. Transmission Potential of Aβ Pathology

It is well known that Aβ protein, representing the most common misfolded peptide in the ageing brain, is deposited in brain parenchyma in AD cases and in cerebral blood vessels, leading to the development of CAA. Among other pathologically aggregated proteins with seeding and propagation properties termed proteopathic seeds, Aβ has proven to be transmissible experimentally both in vitro and in animal models [30,31,32]. As far as the human transmission potential of Aβ seeds is concerned, first Jaunmuktane et al. [33], having studied a group of young iCJD patients after treatment with hGH from cadaveric pituitary glands, observed substantial parenchymal and vascular Aβ deposition in half of the examined individuals in absence of known pathogenic mutations. Consistent with typical AD pathological features in conjunction with CAA histological characteristics, the aforementioned findings were indicative of iatrogenic transmission of Aβ pathology in addition to CJD in recipients of cadaveric hGH, raising concerns regarding the transmissibility of AD. Interestingly, Aβ and prion pathology seem to develop independently, as Aβ seeds did not co-localize with prions among the examined affected subjects. Subsequently, the treatment-induced transmission of Aβ seeds in parallel with iCJD was experimentally confirmed by Purro et al. [34].

In an attempt to further explore the transmission potential of Aβ seeds through other known iatrogenic exposure routes of prions, Frontzek et al. [35] investigated a group of young iCJD patients after neurosurgery with dural grafting. The researchers observed both brain parenchymal Aβ plaques and CAA with high frequency in cases of iCJD after dural grafting in absence of a family history of early-onset dementia, suggesting that, like pituitary extracts, hDM grafts are able to elicit Aβ pathology with prolonged incubation periods. Similarly, Kovacs et al. [36] reported the presence of Aβ deposits in the brain of iCJD patients who underwent hDM implantation through neurosurgery, suggesting that hDM may act as a potential source of Aβ seeds. However, it was pointed out that seeding properties of Aβ are not coupled with the development of the complete clinicopathological phenotype of AD, implying that transmitted Aβ pathology may not be able to induce relevant clinical symptoms.

Consistent with previously mentioned data, Hamaguchi et al. [37] found a significant association of cadaveric DM grafting with subpial Aβ deposition and amyloid angiopathy in iCJD patients. When compared to sporadic CJD cases, patients with iCJD are accompanied by an increased risk of harboring Aβ pathology at a relatively young age with a clinical phenotype of CAA, rather than AD, according to Cali et al. [7]. The researchers concluded that the Aβ pathology of iCJD is clearly distinguished from that of typical AD.

Apart from that, Ritchie et al. [38] demonstrated the seeding properties of Aβ independently of the prion accumulation in hGH recipients, strongly arguing for Aβ deposition potential in the absence of prion pathology and, thus, raising the possibility of treatment-induced transmission of both CAA and AD.

#### 3.2.2. Iatrogenic Cerebral Amyloid Angiopathy (iCAA)

CAA represents a neurological disorder that reflects the progressive accumulation of Aβ protein and subsequent deposition of amyloid plaques in the walls of cortical and leptomeningeal vessels, resulting in vessel damage and impairment of normal blood flow [39]. It is well documented that the aforementioned vascular alterations parallel with both significant hemorrhagic and ischemic risk and correlate well with MCI and dementia [40,41]. Emerging with a wide clinical spectrum of cognitive decline, seizure, headache, and confusion, as well as language or speech difficulties, a sporadic form of CAA commonly presents with an increased incidence rate among individuals of advanced age [39].

Until recently, two types of CAA have been identified: sporadic, which is characterized by an age-related character, and hereditary, which develops because of pathogenic mutations. It has only been in the last few years that an acquired form of CAA has been newly described. It is of particular interest that in a setting of a neurocognitive disorder among more than half of the individuals over 80 years, sporadic CAA is clinically evident [42]. Particularly within a population of AD patients, signs of CAA have been demonstrated in the vast majority of affected individuals, suggesting the significantly unfavorable impact of CAA on cognitive function [39,42]. Cognitive impairment observed in patients with CAA may exhibit a gradual, stepped or rapidly progressive character depending on the clinical phenotype of the disease. More specifically, CAA with clinical manifestations of chronic ischemic leukoencephalopathy, lobar microhemorrhages and lobar lacunae is accompanied by a gradual type of cognitive decline, while an AD diagnosis may coexist. Furthermore, in cases of recurrent lobar hemorrhages as a result of underlying CAA, stepped cognitive impairment emerges, while a rather rapidly progressive cognitive decline may present secondary to inflammatory variants of CAA [43,44].

Recently, several cases of iatrogenic CAA have been increasingly reported. It is well established that some aspects of Aβ pathology may carry a transmission potential in certain circumstances, including administration of hGH as well as transplantation of hDM grafts regardless of iCJD occurrence, suggesting that the subsequent development of CAA may be treatment-induced through transmission routes similar to those of prion diseases. In an attempt to explore the potential human-to-human transmission of CAA through neurosurgery, Jaunmuktane et al. [45] reported a group of patients with a past medical history of neurosurgical intervention during childhood or teenage years in absence of predisposing genetic risk factors that presented with intracerebral hemorrhage approximately three decades later as a result of severe young-onset CAA. Thus, it was concluded that CAA may be transmitted by surgical instruments carrying traces of misfolded Aβ protein. Additional clinical cases of iatrogenic early-onset CAA have been identified in a setting of exposure to dura mater (by grafting or embolization) some decades earlier, highlighting the role of contaminated dura as the source of misfolded Aβ protein [46,47].

With regards to the transmission route mediated by blood transfusion, Zhao et al. [48] explored the linkage between the risk of intracerebral hemorrhage among donors and recipients of red blood cell products. Noteworthy, it was observed that donation of red blood cells from individuals who later developed multiple spontaneous intracerebral hemorrhages was coupled with a significantly increased risk of subsequent intracerebral hemorrhages among recipients, raising the possibility of Aβ transmission and, as a result, occurrence of CAA via blood transfusion. Interestingly, there was also higher risk of intracerebral hemorrhages among recipients from donors with one intracerebral hemorrhage and diagnosis of dementia. Of great clinical relevance, Kaushik et al. [49] reported two cases presented with severe CAA at a relatively young age with a history of blood transfusion in infancy and in absence of relevant genetic risk factors, suggesting that exposure to blood transfusion may serve as a seeding event and induce CAA. Similarly, DiFrancesco et al. [50] identified a group of patients with iCAA and a history of past blood transfusion. Although in all cases blood transfusion coincided with neurosurgery and/or dura transplant, the researchers demonstrated that the development of CAA in a setting of a previous blood transfusion might be plausible.

Highly indicative of the emergence of iCAA as a newly recognized clinical concept is the constantly increasing number of reported cases. Several individual case reports and case series published in the literature have been noting the iatrogenic transmission potential of CAA, thus further elucidating the potential exposure routes, the clinical profile and the neuropathological features of affected subjects [51,52,53,54]. In a case series and review of cases from the literature, Banerjee et al. [55] aimed to explore natural history and the wide clinical spectrum of treatment-induced CAA. In line with previously reported cases, all patients with iCAA experienced a previous medical or surgical procedure and presented with early-onset recurrent intracerebral hemorrhages, seizures and rapidly progressive cognitive impairment three to four decades later. It is of particular interest that iatrogenic CAA cases showed a male predominance, while no correlation was found with pathogenetic mutations or *apolipoprotein E* (*ApoE*) genotype. As far as genetic factors are concerned, it has been postulated that they play only a minor role in the development of iCAA among young individuals with an *ApoE* E3/E3 phenotype in absence of relevant pathogenic mutations being mostly present in iCAA cases [56]. Taken all together, diagnostic criteria for iCAA have been established, including a young age of onset below 55, a history of potential exposure involving cadaveric human CNS tissues or a relevant neurosurgical procedure, radiological and clinical features compatible with CAA, while evidence indicative of Aβ accumulation within the CNS in conjunction with absence of genetic risk factors for amyloidosis are essential along the diagnostic approach to rule out alternative potential causes of CAA [55].

According to a recently published large study [57], almost 50% of young patients with CAA were suspected of having iCAA with a mean age at presentation of 43 years and latency between the potential exposure and disease onset of 36 years. Consistent with previously published data, the most frequent presenting symptoms of iCAA comprised intracerebral hemorrhages, transient focal neurological episodes and seizures, with a higher proportion of men reported within the group of patients under evaluation. Interestingly, 20% of the patients exhibited a degree of cognitive decline during follow-up.

It is well documented that iCAA should be considered in young, affected individuals with a history of potential exposure to exogenous Aβ aggregates with typical symptom onset at an age under 55 years considered “too young” for sporadic CAA. However, the age spectrum of the disease seems to be much wider, as recently, cases of possible iatrogenic CAA in adults aged 65 years and older presenting in later life have been reported [58]. Taking into consideration the proposed diagnostic criteria, five cases of possible iatrogenic CAA have been identified. All affected individuals presented at an age over 65 years and had undergone a neurosurgical procedure with a high probability of cadaveric dural grafting utilization. It should be noted that in all cases, the latency between the potentially responsible for exposure procedures and clinical presentation ranges within three and four decades, that is typical among iCAA patients. Although the clinical phenotype appears to be distinct from that observed in younger onset iCAA cases with cognitive impairment and transient focal neurological episodes being mainly present, the diagnostic approach of iCAA within an older age group constitutes a significant challenge with genetic studies and evidence indicative of Aβ accumulation within the CNS being of less relevance.

#### 3.2.3. Iatrogenic Alzheimer’s Disease (iAD)

Alzheimer’s disease represents the most common neurodegenerative disease worldwide with an unfavorable impact on memory and cognitive function of affected subjects. Emerging with a high incidence rate among patients above 65 years, around two-thirds of total dementia cases in older individuals are attributed to AD development [59].

Like human prion diseases, AD is characterized by the accumulation and spread of specific misfolded proteins within the brain. Aβ deposition combined with the formation of neurofibrillary tangles composed of hyperphosphorylated tau proteins in the brain parenchyma constitute the pathological hallmarks of AD, required in order for a diagnosis of AD to be assigned [60]. Although both abnormal proteins are of paramount importance for the emergence of symptomatic disease, genetic and biomarker evidence suggests that the cerebral accumulation of misfolded Aβ is capable of triggering the pathological cascade of events, ultimately leading to progressive loss of episodic memory and diverse cognitive deficits [61]. Nevertheless, tau deposition seems to parallel better with cognitive decline and neuronal loss [62]. Compared to human prion diseases, AD mainly manifests with a rather progressive cognitive decline [36].

The human-to-human transmission potential of exogenous Aβ aggregates resulting in the clinical manifestation of iCAA after a long incubation period posed significant concerns regarding the transmissibility of AD and raised the possibility that AD may be characterized by prion-like properties as well, eventually developing in individuals exposed to known transmission routes at even longer latency. Although iatrogenic transmission of Aβ pathology has been well documented among affected individuals regardless of coexistent iCJD, no evidence of substantial tau pathology has been observed in patients with iCJD, treatment-induced Aβ pathology or both [62,63]. According to recent investigations in humans, exposure to Aβ seeds in a clinical setting can lead to Aβ deposition, mainly manifested as CAA [55], which after prolonged incubation periods seems to be coupled with substantial tau pathology [63]. Until now, however, it has not been clear whether Aβ seeds can induce the full pathologic and cognitive phenotype of AD [64].

Recently, Jaunmuktane et al. [63] demonstrated that significant tau pathology, indistinguishable from AD type changes, can occur in a setting of iatrogenically transmitted Aβ pathology after particularly long incubation periods, shedding light on the temporal development of tau pathology. At least 35 years seems to be a prerequisite for neuropathological alterations compatible with AD diagnosis to occur. Regarding the clinical profile of the examined patients, only one out of three reported a history of cognitive decline, while two out of three presented with no degree of cognitive impairment, neither at the time of biopsy nor during follow-up. In line with previously published data, Milani et al. [65] identified a patient with a prior neurosurgical procedure who presented with combined vascular and parenchymal Aβ pathology as well as neurofibrillary changes consistent with AD findings after a prolonged incubation period, further expanding the already existing knowledge regarding the acquired form of AD. Nonetheless, considerable tau pathology is not evident in all cases reported to date with similarly lengthy latencies [35,36,46].

Inconsistent with previously published data regarding the inefficiency of Aβ pathology alone to induce the typical phenotype of AD [36], Banerjee et al. [66] explored the possibility that childhood treatment with cadaveric hGH may eventually lead to AD and reported a group of cadaveric hGH recipients who presented either with early-onset cognitive decline and/or biomarker findings highly indicative of AD. More specifically, dementia affecting the performance of daily activities was observed with a latency of three to four decades among most young individuals examined with a history of prior hGH administration, while MCI predominantly affecting the patient’s personality and behavior, subjective cognitive symptoms only, or complete absence of symptoms were reported among the remaining individuals. Furthermore, Aβ deposition in parenchyma together with neurofibrillary tangles of tau have been demonstrated. Thus, the clinical profile and the biomarker changes within the phenotypic spectrum of AD seemed to reflect Aβ transmission from contaminated hGH received in childhood. Notably, the combination of unusually young age at onset, atypical presentations with early involvement of multiple cognitive domains (nonmemory and noncognitive), as well as the absence of one or recurrent intracerebral hemorrhages and brain imaging biomarkers compatible with CAA might be of added value in differentiating this potentially iatrogenic variant of AD from CAA and other AD types.

Similarly, Hernández-Fernández et al. [67] identified an additional case of iCAA that fulfilled criteria for comorbidity with AD and presented with a rather atypical clinical phenotype of AD with multi-domain involvement at a young age and evident additional neurological signs including progressive ataxia or episodes of encephalopathy, while prominent hemorrhagic manifestations and relevant imaging biomarkers were lacking. The researchers concluded that the aforementioned case reflected the iatrogenically transmitted co-pathology of CAA and AD with distinct clinical manifestations and biomarkers from those seen both in iCAA and sporadic AD.

In an attempt to mitigate a potential public health crisis resulting from the implications of the study conducted by Banerjee et al. [66], Nath et al. [68] further examined the total of cases reported in order to evaluate whether they support a diagnosis of AD. Taking into account that an AD diagnosis in a clinical setting of progressive cognitive decline requires demonstration of either amyloid and tau pathology or amyloid and tau biomarkers, the researchers concluded that most cases do not fulfill the necessary criteria for AD diagnosis according to well-established clinical and pathological standards. Thus, it was pointed out that none of the above cases should be assigned a definitive AD diagnosis and alternative diagnoses should instead be considered.

It is of particular interest that Singh et al. [69] recently demonstrated a transplantable variant of AD in a preclinical context. More specifically, pathological hallmarks of AD were observed among healthy recipients in a setting of stem cell transplantation from donors harboring a pathogenic mutant allele, leading ultimately to the development of cognitive impairment. The conclusion of this investigation argues for potential iatrogenic transmission in AD patients. Further to this, taking into consideration the close brain environment of *primates* to that of humans due to their phylogenetic proximity, Lam et al. [70] studied a group of AD-inoculated *primates* in search of neuropathological alterations and clinical manifestations indicative of AD. It was demonstrated that Aβ and tau pathologies may be induced in non-human *primates* after being inoculated with human AD brain extracts. The developed neuropathologic alterations were associated with cognitive deficits and bilateral cerebral atrophy, thus suggesting the transmission potential of an AD-like phenotype within a group of *primates*. Whether Aβ and tau iatrogenic transmission can be translated into clinical manifestations among humans remains to be elucidated.

## 4. Discussion

An extensive literature review was conducted in order to elucidate the transmission potential of dementia in a clinical setting of a neurodegenerative disease. Original articles dealing with the underlying pathomechanisms and the distinct clinical phenotype of acquired type of cognitive decline were identified and reviewed.

Proteinopathies represent a group of diseases characterized by the deposition of misfolded protein aggregates. These abnormally accumulated proteinaceous structures in the brain can trigger the pathological cascade that leads to neurologic dysfunction and subsequently to clinically evident neurodegeneration with manifestations within the dementia spectrum. Prion proteins, Aβ and tau being among them, stand for the pathological hallmarks of various neurodegenerative diseases, reflecting the main underlying pathomechanisms [62].

In an attempt to explore the underlying molecular pathways involved in iatrogenically triggered amyloidogenesis, Bonilauri [71] proposed a classification of relevant cases according to the nature of medical intervention and subsequent impact on amyloid proteins. As far as neurodegenerative disorders including AD, CAA and CJD are concerned, the amyloidogenic cascade can be initiated by the transmission of amyloid oligomers and fibrils through different medical procedures. Already found in a pre-existing amyloid state in cases of organ/tissue transplantation or contaminated instruments, amyloid seeds may enhance and ultimately accelerate the formation of insoluble mature amyloid fibrils, thus disrupting normal tissue functionality and leading to a wide range of clinical manifestations. According to growing experimental evidence, the ability of misfolded proteins to continuously induce the conversion of similar physiological proteins into a pathological form through seeding is a characteristic feature not only of prion protein but also Aβ, tau and a-synuclein [10]. A common molecular mechanism seems to be responsible for the replication and spread of these different misfolded protein aggregates within the central nervous system [3].

It is of great interest that a-synuclein might exhibit at least some of the misfolded protein properties, including the generation of a-synuclein aggregates that act as a template for the formation of seeds, propagation between cells and spread across certain anatomical pathways. As a result, both neurons and glial cells are affected and distinct clinical phenotypes may emerge, Parkinson’s disease (PD) being among them. Although human transmission of a-synuclein pathology has not yet been experimentally proven, a-synucleinopathies coupled with various clinical manifestations may follow a similar iatrogenic exposure route, raising awareness regarding potential iatrogenic PD cases [3,10].

It is well established that human prion diseases carry a substantial transmission potential between individuals after a prolonged incubation period with a few potential exposure sources having been identified up to today. Documented transmission routes in humans may originate commonly from cadaveric-derived dural graft or embolization material implantation in a neurosurgery setting, cadaveric human growth hormone administration and utilization of contaminated neurosurgical instruments [3,6]. Medical procedures responsible for disease transmission do not only refer to brain- or spinal cord-related interventions, as additional surgical procedures requiring the use of cadaveric dura tissue are considered potential transmission sources as well [56].

According to an increasing body of evidence, other disease-specific misfolded proteins may exhibit prion-like qualities in terms of propagation and spread along neuroanatomical pathways in the brain, thus raising the question of whether these abnormal forms of proteins demonstrate transmission potential via similar iatrogenic sources of infection. Recently accumulated data are highly indicative of the fact that prion disease may serve as a model for various neurodegenerative diseases, reflecting prion hypothesis as the shared underlying pathomechanism [6]. More specifically, transmission of Aβ pathology was initially observed based on the neuropathological findings in cases of iCJD [7,33,35,36] and only later also in the absence of it among young individuals with early onset CAA combined with a history of neurosurgery or other invasive medical procedure [37,38,45,46,47].

Of great concern emerges the possibility that common neurodegenerative and neurovascular conditions related to AD and CAA might be transmissible under certain conditions through similar routes of exposure. Regarding iatrogenic CAA, more than 70 cases have been reported worldwide with the great majority being attributed to the administration of contaminated cadaveric human materials derived from the central nervous system [54,55,57]. Despite the absence of their external validation, diagnostic criteria for iCAA have been recently proposed in an attempt to develop an accurate diagnostic framework and further facilitate the diagnostic approach in cases of individuals at risk of acquired CAA. Combining young age at symptom onset < 55, a history of potential exposure through treatment using cadaveric human CNS tissues or relevant neurosurgical procedures, and both clinical and radiological manifestations in conjunction with evidence of Aβ deposition in the CNS and exclusion of hereditary causes of amyloidosis may significantly enhance the diagnostic accuracy of iCAA [55,72].

Depending on the age at exposure and the latency of clinical and radiological manifestations, patients may present at a relatively young age [56]. However, considering the long incubation period between exposure and symptom onset, typically ranging between three and four decades [55,57], iCAA rarely occurs below the age of 30 years. Furthermore, several cases of possible iCAA have been identified in adults aged 65 and older, all of whom reported a history of neurosurgical procedure [58], suggesting that individuals might also be affected at an advanced stage of life, further widening the age range of the disease [56]. Although iCAA is usually considered in patients with early onset of symptoms and a compatible history of potential exposure, the differential diagnosis between iCAA and sCAA in older individuals remains a great challenge. Notably, a distinct clinical profile of iCAA has been observed among patients with higher rates of intracranial hemorrhage recurrence, seizures and rapid cognitive decline, when compared to sCAA [55]. Progressive cognitive impairment represents a less common presenting feature in a setting of sCAA [55].

Of significant clinical importance is the fact that the neuropathological profile observed in individuals exposed to exogenous Aβ aggregates seems to be distinguishable from that typically found in AD cases [62]. Recent studies indicate that Aβ deposition in the brain parenchyma and blood vessel walls following iatrogenic transmission is coupled with either minimal or no tau pathology. Patients with iCJD associated both with hGH treatment and dural grafting were found to exhibit no substantial tau pathology, thus differentiating the Aβ proteinopathy in iCJD from AD [35,36,63]. Unlike prion diseases, the full clinical phenotype of AD has not been reproduced in these individuals; in particular, there is a notable absence of progressive cognitive decline [62]. In contrast with data previously reported, it was demonstrated that particularly lengthy incubation periods between potential exposure to Aβ aggregates and disease onset may represent a prerequisite for the development of significant tau pathology in addition to CAA and parenchymal Aβ accumulation [63]. Coupled with broader clinical implications in terms of neurosurgical practices and primary prevention strategies aimed at effectively mitigating the potential transmission risk, recent evidence suggests that AD may exhibit a human-to-human transmission potential in certain circumstances [66]. Nevertheless, results of further studies are not in line with the aforementioned data [68].

Despite the rarity of human prion diseases, their transmission potential, prolonged incubation periods and notable resistance to conventional decontamination methods raise considerable concerns to public health. Up to today, the implementation of improved decontamination approaches regarding surgical instruments and biological products in conjunction with enhanced identification of potentially infected individuals has led to a significant decrease of CJD inter-human transmission risk. However, optimal diagnostic tools able to detect low levels of various misfolded proteins in human-derived materials or surgical instruments are still lacking. The development of an accurate screening test capable of being utilized at a preclinical stage may further mitigate the iatrogenic transmission risk.

Although only occasional cases of iCJD with exceptionally prolonged latencies are still appearing following the suspension of cadaveric hDM and hGH administration, it can be postulated that further cases of iCAA will be identified soon due to the significantly longer incubation periods observed. Thus, awareness should be raised in cases presenting with clinical and neuroimaging features consistent with environmentally acquired CAA. Currently, researchers are concerned that healthy individuals with a history of potential exposure to Aβ seeds may be at risk of iatrogenic AD.

Given that long-term follow-up data of patients with misfolded protein-related disorders may prove of paramount importance in terms of understanding the underlying mechanisms driving disease progression, a growing need is emerging for both regular clinical and neuroimaging follow-up of affected individuals in an attempt to further elucidate the natural course of the relevant disorders and facilitate the implemented diagnostic approach. Furthermore, international registries and large-scale epidemiological studies are crucial to explore additional cases of iatrogenically transmitted causes of dementia, since they occur rarely when compared to sporadic and hereditary cases.

## 5. Conclusions

The present review summarizes all the up-to-date evidence regarding the newly recognized, rather intriguing concept of iatrogenic dementia. Disorders associated with misfolded protein deposition in the brain and potentially manifested with cognitive decline, encompassing Alzheimer’s disease, Creutzfeldt–Jakob disease and cerebral amyloid angiopathy, seem to share common underlying pathomechanisms, thus suggesting that they exhibit similar inter-human transmission potential through well documented transmission routes. Distinguishable from both sporadic and hereditary forms, environmentally acquired dementia originating from prion or Aβ-related disorders pose significant public health concerns with emerging broad implications. Although representing rare entities, increased awareness in the identification of potential iatrogenic cases of cognitive impairment remains essential for targeted management.

## Figures and Tables

**Figure 1 biomolecules-15-00522-f001:**
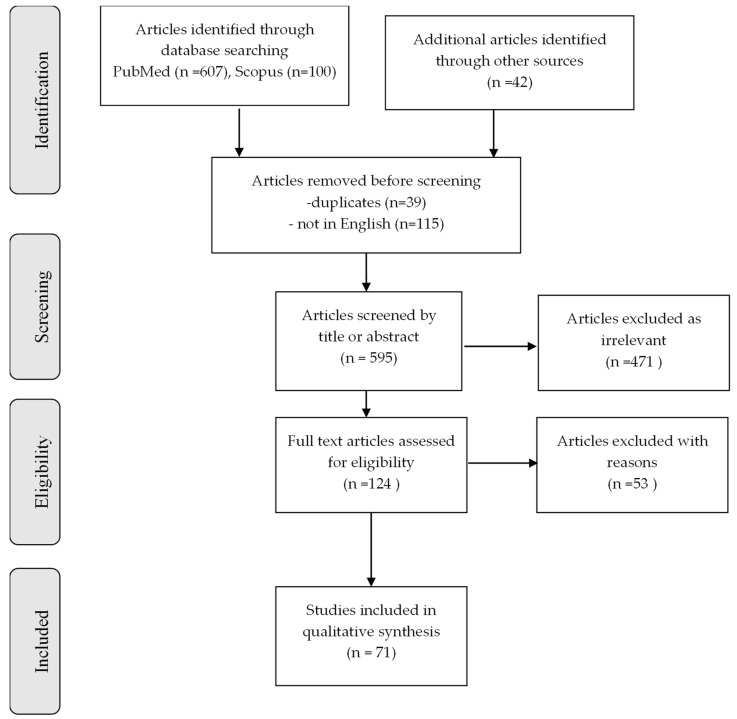
Study flow chart (PRISMA diagram).

## Data Availability

Not applicable.

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
