# Peer review of "Iatrogenic Dementia: Providing Insight into Transmissible Subtype of Alzheimer’s Disease, Creutzfeldt–Jakob Disease and Cerebral Amyloid Angiopathy"

_biomolecules, 2025, doi:10.3390/biom15040522_

Round 1

Reviewer 1 Report

Comments and Suggestions for Authors

The authors carefully described the accumulating evidence, along many years, of the iatrogenic origin of misfolding protein related neurodegenerative disorder.

The authors performed a wide description of documented transmission routes in humans that may originate commonly from cadaveric-derived dural graft or embolization material implantation in a neurosurgery setting, cadaveric human growth hormone administration and utilization of contaminated neurosurgical instruments. They posed significant concerns regarding the transmissibility of AD and raised the possibility that AD may be characterized by prion-like properties as well, leading to common neurodegenerative diseases or amyloid angiopathy. This review raised the importance of developing an accurate screening test to mitigate the iatrogenic transmission risk and raising awareness of the identification of potential iatrogenic cases showing clinical and neuroimaging features consistent with environmentally acquired proteinopathies.

Major issues:

Although this review address a specific aim, it would be useful to mention at least in the discussion, that strong data also suggest that misfolded human α-synuclein can be taken up by neurons and glial cells, amplifying and corrupting endogenous α-synuclein through templated misfolding. This transmission across neural systems to spread throughout the brain can follow the same potential exposure route, expanding the above-mentioned awareness also to Parkinson’s diseases.

Minor issue:

Reevaluation of english grammar.

Comments on the Quality of English Language

I found several typing as well English grammar mistakes. Just in the introduction, for example:

line 39: by the year 2030 and 2050 - by the years 2030 and 2050 

line 48: that are subsequently being accumulated - that are subsequently  accumulated  

line 55: proteins, namely including amyloid-beta - proteins, including amyloid-beta

line 60: cells and spread across - cells and spreads across 

line 65: potential with prolonged incubation - potential with a prolonged incubation  

line 66: exposure and onset - exposure and the onset  

line 67: inherited subtype, account only - inherited subtypes, accounts only 

line 80: valuable insight into - valuable insights into 

and so on...

Author Response

Comments 1: The authors carefully described the accumulating evidence, along many years, of the iatrogenic origin of misfolding protein related neurodegenerative disorder. The authors performed a wide description of documented transmission routes in humans that may originate commonly from cadaveric-derived dural graft or embolization material implantation in a neurosurgery setting, cadaveric human growth hormone administration and utilization of contaminated neurosurgical instruments. They posed significant concerns regarding the transmissibility of AD and raised the possibility that AD may be characterized by prion-like properties as well, leading to common neurodegenerative diseases or amyloid angiopathy. This review raised the importance of developing an accurate screening test to mitigate the iatrogenic transmission risk and raising awareness of the identification of potential iatrogenic cases showing clinical and neuroimaging features consistent with environmentally acquired proteinopathies.

Major issues: Although this review address a specific aim, it would be useful to mention at least in the discussion, that strong data also suggest that misfolded human α-synuclein can be taken up by neurons and glial cells, amplifying and corrupting endogenous α-synuclein through templated misfolding. This transmission across neural systems to spread throughout the brain can follow the same potential exposure route, expanding the above-mentioned awareness also to Parkinson’s diseases.

Response 1: Thank you for pointing this out. We agree with this comment. Therefore, we have added some additional data regarding the potential transmissibility of Parkinson's disease. The above change can be found in page 10, line 446-460, within the revised manuscript.

Comment 2: Minor issue: Reevaluation of english grammar. I found several typing as well English grammar mistakes. Just in the introduction, for example:

line 39: by the year 2030 and 2050 - by the years 2030 and 2050 

line 48: that are subsequently being accumulated - that are subsequently  accumulated  

line 55: proteins, namely including amyloid-beta - proteins, including amyloid-beta

line 60: cells and spread across - cells and spreads across 

line 65: potential with prolonged incubation - potential with prolonged incubation  

line 66: exposure and onset - exposure and the onset  

line 67: inherited subtype, account only - inherited subtypesaccounts only 

line 80: valuable insight into - valuable insights into 

Response 2: We agree with this comment. We have, accordingly, revised our manuscript in order to correct some typing and English grammar mistakes found. All of the above mentioned by the reviewer mistakes have been corrected and english grammar has been reevaluated throughout the original manuscript. The revisions have been marked in red. 

Reviewer 2 Report

Comments and Suggestions for Authors

Dear authors, I like your paper. It is very due and highlighted the emerging problems connected with prion-like behavior of several proteins involved in AD, CAA and other forms of neurodegeneration. It is of interest that different proteins explore probably the same routes of transmission in the human brain. You mentioned that monkeys may be infected by human material and mentioned several other ways of transmission. There are hundreds of animal models of AD and other neurodegenerative diseases. I think it is easier to study transmission routes of beta-amyloid etc. in these model organisms than in humans but unfortunately you did not mention a single paper describing such cases. Please, do it if there is any.

Author Response

Comment: Dear authors, I like your paper. It is very due and highlighted the emerging problems connected with prion-like behavior of several proteins involved in AD, CAA and other forms of neurodegeneration. It is of interest that different proteins explore probably the same routes of transmission in the human brain. You mentioned that monkeys may be infected by human material and mentioned several other ways of transmission. There are hundreds of animal models of AD and other neurodegenerative diseases. I think it is easier to study transmission routes of beta-amyloid etc. in these model organisms than in humans but unfortunately you did not mention a single paper describing such cases. Please, do it if there is any.

Response: Thank you for pointing this out. Our original manuscript includes a study by Lam et al. which was conducted in a group of AD-inoculated primates, highlighting the transmission potential of an AD-like phenotype in such animal models. Taken that our research was mainly focused on humans and the potential human-to-human transmissibility of specific neurodegenerative diseases, we haven't further modified our manuscript in this direction.